**Data Availability Statement:** All relevant data are within the manuscript and its Supporting Information files.

**Funding:** The authors received no specific funding for this work.

# Acetone application for administration of bioactive substances has no negative effects on longevity, fitness, and sexual communication in a parasitic wasp

**Anne-Sophie Jatsch** ⓘ, **Joachim Ruther** ⓘ *

Institute for Zoology, University of Regensburg, Regensburg, Germany

* joachim.ruther@ur.de

## Abstract

Administration of defined amounts of bioactive substances is a perseverative problem in physiological studies on insects. Apart from feeding and injection, topical application of solutions of the chemicals is most commonly used for this purpose. The solvents used should be non-toxic and have least possible effects on the studied parameters. Acetone is widely used for administration of chemical substances to insects, but possible side-effects of acetone application on fitness and behavioral parameters have been rarely investigated. Here we study the effects of acetone application (207 nl) on fitness and sexual communication in the parasitic wasp *Nasonia giraulti* Darling. Application of acetone had neither negative effects on longevity nor on offspring number and offspring sex ratio of treated wasps. Treatment of females hampered courtship and mating of *N. giraulti* couples neither directly after application nor one day after. Male sex pheromone titers were not influenced by acetone treatment. Three application examples demonstrate that topical acetone application is capable of bringing active amounts of insect hormones, neuromodulators, and biosynthetic precursors even in tiny insects. We advocate the use of acetone as a convenient, conservative, and broadly applicable vehicle for studying the effects of bioactive substances in insects.

## Introduction

Insects are essential for the stability of terrestrial and aquatic ecosystems and impact human life in many ways [1, 2]. As pollinators, for instance, insects are crucial for the reproduction of numerous plant species and ensure the availability of fruits for human nutrition [3]. As herbivores, they compete with us for all kinds of plant-derived products [4]. Carnivorous insects control herbivore populations [5] and many insects are vectors of diseases threatening human health [6]. Given their enormous ecological and economic importance, all aspects of insects' biology are investigated extensively by scientists. A perseverative problem in this research is getting bioactive chemical compounds into the insect to investigate their effects on the treated individuals. The method by which this is achieved should allow exact control of the applied

**Competing interests:** The authors have declared that no competing interests exist.

dosage and have as few side-effects as possible to enable assignment of the observed effects to the respective chemical. Methods most commonly used to bring chemical substances into insects include feeding, injection, and topical application of the chemical under investigation [7]. When choosing a suitable method, it has to be considered that the way by which bioactive chemicals are administered can influence the outcome of the experiments [7, 8]. Administration by feeding requires the compounds of interest to be soluble in water to enable voluntary uptake by the insects. Water-soluble solvents may be used as solubilizers and the addition of feeding stimulants such as sucrose may increase the motivation of insects to feed on the offered solutions of often unpalatable compounds [7, 9–12]. While the concentration of an active compound in the provided solution can be exactly adjusted, the amounts consumed by the insect is often difficult to control. Hence, in studies, in which the dosage of the studied compound is critical, administration by feeding is often suboptimal.

Injection of the active compound using commercially available or custom-made microinjection systems can circumvent this problem [7, 8, 10, 11, 13, 14]. Penetration of the insect's integument, however, means a drastic intervention and bears the risk of undesired side-effects due to, for instance, the permeation of pathogens or the injury of internal organs.

Application of the active compound dissolved in a solvent is a convenient and non-invasive alternative to microinjection techniques. This approach has been frequently used in insect studies investigating, for instance, the effects of insect hormones and hormone analogues [15–17], neurotransmitters and -modulators [7, 18], essential oils [19, 20], synthetic insecticides [21–23], or the metabolism of biosynthetic precursors [24–26]. One prerequisite for meaningful results is the availability of a suitable solvent. This should dissolve chemicals of a broad polarity range, not react with the active compound under investigation, possess low or no toxicity at the applied dosage, and penetrate quantitatively into the treated insect. Frequently used solvents in insect studies include alcohols such as ethanol or methanol [27, 28] which are relatively polar and thus not well suited to dissolve non-polar active compounds and to pass the lipid layer covering the insect cuticle. Furthermore, orally administered ethanol suppressed learning behavior in honeybees [29] and alcoholic solvents fostered enzymatic transesterification of the active compound when used to inject juvenile hormone into locusts [30].

Short-chain alkanes are well suitable to dissolve non-polar substances. However, alkanes such as hexane and alkane mixtures such as kerosene, consisting of alkanes below decane, have a knock-down effect on many insects [31, 32] and are thus poorly suitable for application purposes. The same is true for aromatic alkanes such as benzene, toluene, or xylene which cause oxidative stress and mortality in exposed insects [33, 34]. Dimethylsulfoxide (DMSO) is a more polar solvent which is considered to have a relatively low toxicity and has been used for the application of more polar chemicals to insects [25, 35]. DMSO, however, is a poor solvent to dissolve non-polar compounds and has been shown to eliminate the response of sensory neurons of insect mechanoreceptors [36]. Furthermore, DMSO caused significant toxic effects when added at concentrations >0.3% to the rearing medium of *Drosophila melanogaster* larvae [37].

A frequently used solvent in physiological and biochemical investigations on insects is acetone [16, 19, 21, 24, 38], a mid-polar solvent dissolving chemicals of a broad polarity range. Acetone has been assessed to possess a relatively low acute toxicity for mammals and aquatic as well as terrestrial arthropods [39]. When applied to insects, acetone did not have any significant effects on insect mortality [32, 40], while exposure to relatively high concentrations of acetone vapor killed eggs, larvae and adults of some stored product infesting insects [40]. To the best of our knowledge, however, subtler sublethal effects of acetone application on fitness-relevant parameters of insects such as fecundity or sexual communication have not been studied so far.

During the past decades, the parasitic wasp genus *Nasonia* (Hymenoptera: Pteromalidae) has become an extensively studied model system for all aspects of parasitic wasp biology [41–45]. The genus consists of the four species *N. vitripennis*, *N. giraulti*, *N. longicornis*, and *N. oneida*. All species are gregarious and parasitize pupae of cyclorraphous flies [44, 46, 47]. The tiny *Nasonia* wasps (length ca. 2 mm) are easy to rear in the lab and have short generation times. This and the availability of sequenced genomes makes them ideal model organisms. Sexual communication in *Nasonia* species is mediated at different levels by pheromones [42]. Males produce in their abdomen hydroxylated lactones and 4-methylquinazoline as substrate-borne, abdominal sex pheromones to attract virgin females. All species produce (*4R*,*5S*)-5-hydroxy-4-decanolide (RS-HDL) while *N. vitripennis* uses (*4R*,*5R*)-5-hydroxy-4-decanolide (RR-HDL) as an additional pheromone component [48–51]. Both lactones are derived from fatty acid metabolism. $^{13}$C labeling experiments have revealed that linoleic acid is a key intermediate in the pheromone biosynthesis of *N. vitripennis* while the other *Nasonia* species have not yet been studied in this respect [52–54]. Another aspect of the pheromone biosynthesis in *Nasonia* that has not yet been studied is the role of insect hormones such as juvenile hormone (JH) and 20-hydroxyecdysone (20-OH-E) which have been found to trigger pheromone biosynthesis in several other insect taxa [13, 14, 17, 55–57].

After contact with a female, males display a stereotypic courtship behavior which is elicited by female-derived cuticular hydrocarbons [58–60]. Male courtship includes the release of oral aphrodisiac pheromones that make females receptive and unresponsive to the males' abdominal sex attractants [61–63]. In *N. vitripennis*, this behavioral switch in females is modulated by dopamine [10]. Several previous studies on the pheromone communication of *Nasonia* species investigated the effect of bioactive molecules on the pheromone communication. The compounds under investigation were administered by feeding [9, 64], injection [10, 50], or application as acetone solutions [21, 24, 26]. As for the latter, a volume of ca. 200 nl was routinely used, and the wasps appeared to be fully vital within one hour after the application. In any of these experiments pure acetone was applied as a control treatment, however, it was unknown so far whether application of the pure solvent has any impact on the fitness and the behavior of *Nasonia* wasps.

In the present study, we use *N. giraulti* as a model species to investigate whether the application of acetone influences relevant fitness parameters of the treated wasps. We furthermore study whether acetone application per se influences the production of the major sex pheromone component RS-HDL in males as well as courtship behavior and mating rate. Finally, we present three application examples of the technique to demonstrate that the application of acetone solutions is a convenient and effective way to study the physiological and biochemical effects of bioactive molecules even in tiny insects. These application examples investigate the effect of JH and 20-OH-E on male RS-HDL titers and the role of linoleic acid as precursor of RS-HDL in *N. giraulti* as well as the role of dopamine as mediator of olfactory plasticity in the pheromone response of *N. vitripennis* females.

## Materials and methods

### Insects

*Nasonia giraulti* originated from the inbred strain NGVA2 and were kindly provided by Thomas Schmitt (University of Würzburg, Germany). *Nasonia vitripennis* originated from the inbred strain used in our previous pheromone studies [48, 49, 65] and was originally collected near Hamburg in Northern Germany. Wasps were reared on freeze-killed puparia of the green bottle fly *Lucilia caesar* as described elsewhere [58]. To obtain wasps of defined age and mating

status for the experiments, parasitoid pupae were excised from host puparia 1–2 days prior to emergence and kept singly in 1.5 ml microcentrifuge tubes until emergence.

## Acetone application

Prior to acetone application, newly emerged wasps were cold-sedated by keeping their microcentrifuge tubes for 10 min on ice. After putting individual wasps backwards on an ice-cooled cylindrical platform under a stereomicroscope, 207 nl of acetone (99.8%, Rotisol$^{TM}$, Carl Roth GmbH, Karlsruhe, Germany), were applied in three dosages of 69 nl each to the abdominal tips of the immobilized wasps. To this end, we used a Nanoliter 2010 microinjector that was mounted to a micromanipulator (both World Precision Instruments, Friedberg, Germany). The solvent was applied through micro capillaries produced from TW 120–6 glass capillaries (World Precision Instruments) using a PC-10 capillary puller (Narishige International, London, UK). This procedure allowed a virtually quantitative penetration of the solvent through the abdominal intersegmental skin and the anal orifice of the insect (see S1 Video of the application procedure). After the application, wasps were set back into their microcentrifuge tubes and kept there until being used in the experiments. Depending on the experiment, treated wasps were either tested three to four hours (0-d old) after the application or the next day (1-d old). Control wasps were subjected to exactly the same procedure, however, without acetone application.

## Longevity

Acetone-treated and control wasps (n = 30) were kept individually at 25˚C in their microcentrifuge tubes and provided with a pellet of cotton soaked with 30 μl of water as well as with a small drop of honey. The following days it was noted each afternoon whether the wasps were still alive until all wasps had died.

## Mating trials

To investigate whether acetone application to females has an impact on the mating behavior of *N. giraulti* couples, we added one virgin male each to the microcentrifuge tubes of 0- or 1-d old (see 2.2) acetone-treated or control females (n = 25 for each treatment). Subsequently, we observed the mating behavior of the couples under a stereomicroscope. The Observer XT scientific software (version 15, Noldus Information Technology, Wageningen, The Netherlands) was used to record the following behavioral parameters: (a) The time until first mounting, (b) the time from first mounting to the female's receptivity signal (lowering of the antennae/opening of the genital orifice), (c) whether a copulation occurred or not, and if so, (d) the copulation duration. Maximum observation time for this experiment was five minutes. Four couples (two treatment and control couples each) did not start courtship within these five minutes. For these couples, "Time until first mounting" was set to 300 s for statistical analysis.

## Fitness and longevity of mated females

The females obtained from the mating trials were kept individually in Petri dishes (Ø 8,5 cm) and provided with five fresh host pupae per day to enable host feeding and oviposition. Hosts parasitized the days before were left inside the Petri dishes. Those females that lived longer than 14 d in this experiment were transferred to new Petri dishes and provided there with five fresh hosts per day. This prevented the daughters of the females to parasitize the newly provided hosts given that the generation time of *N. giraulti* under rearing conditions is 14 days. Hence, all wasps emerging inside the Petri dishes after the death of the focal females were

offspring of this female. After the death of acetone-treated and control females, the respective Petri dishes were kept for 15 additional days under rearing conditions, until all wasps of the next generation had emerged. After this time, Petri dishes were kept overnight at -20°C and the offspring number of individual females was counted. For the determination of individual offspring sex ratios, 30 randomly chosen wasps per brood were sexed. Sex ratios of individual broods were calculated as proportion of female offspring.

### Effect of acetone application on male pheromone titers

To test whether the application of acetone has an effect on the biosynthesis of the sex pheromone component RS-HDL in *N. giraulti* males, newly emerged males (n = 20) were treated with 207 nl of acetone as described above. Twenty control males were subjected to the same procedure without acetone application. *Nasonia giraulti* males emerge without any pheromone in their pheromone glands and pheromone titers maximize two days after emergence [51]. Hence, we freeze-killed (-20°C) treated and control males after two days, dissected their abdomens with a scalpel, and extracted each abdomen for 30 min in 40 µl of dichloromethane containing 10 ng/µl methyl undecanoate as an internal standard. After removal of the abdomens, pheromone extracts were stored at -20°C until being analyzed by coupled gas chromatography/mass spectrometry (GC/MS, see section Chemical Analyses).

### Application examples

**Effect of insect hormones on pheromone titers.** The ubiquitously occurring insect hormones 20-OH-E and JH are not only controlling molting processes of juvenile insects but, in many insects, influence also pheromone biosynthesis in the adult stage [14, 17, 57]. Here, using the acetone-application method, we investigated how 20-OH-E and JH III affect pheromone biosynthesis in *N. giraulti* males. To this end, we produced solutions of 20-OH-E (Sigma-Aldrich, Saint Louis, MO, USA) and JH III (Sigma-Aldrich) in acetone (5 µg/µl) and applied 207 nl of the solutions (representing a dosage of 1.04 µg or 1040 ng, referred to as JH 1000 and 20-OH-E-1000 in the results) to the abdomens of newly emerged males as described above. Control males were treated with pure acetone (n = 20 for each treatment). Given that we found a significant effect of JH III at this dosage (see Results), we tested this compound again with a tenfold diluted acetone solution of JH III (500 ng/µl, representing a dosage of 103.5 ng, referred to as JH 100 in the results, n = 20). Two days after the treatment, pheromone extracts of individual males were produced as described above and analyzed by GC/MS (see section Chemical Analyses).

**Biosynthesis of the male sex pheromone from linoleic acid.** Linoleic acid is a precursor of male sex pheromone components RR- and RS-HDL in the congeneric species *N. vitripennis*. This has been demonstrated by rearing males on hosts which had been fed a diet experimentally enriched in fully $[^{13}C_{18}]$-linoleic acid [52]. Mass spectra of RR- and RS-HDL from these males indicated the incorporation of $[^{13}C_{18}]$-linoleic acid into the male sex pheromone. Here, we used the acetone application method to investigate whether linoleic acid is also the pheromone precursor in *N. giraulti*. To this end, we applied 207 nl of an acetone solution of $[^{13}C_{18}]$-linoleic acid (10 mg/ml, Campro Scientific, Berlin, Germany) to the abdominal tips of 1-d old *N. giraulti* as described above. The pure solvent was applied to control wasps of the same age (n = 5 for each treatment). The next morning, males were frozen and the pheromone was extracted the next day as described above. Pheromone extracts were analyzed by GC/MS (see section Chemical Analyses) and mass spectra of RS-HDL were checked for the occurrence of the diagnostic ions m/z 90, 107, 120 indicating the incorporation of $^{13}C$ from fully labeled $C_{18}$-fatty acids into the RS-HDL molecule [24, 26].

**Effect of dopamine on the female sex pheromone response in *Nasonia vitripennis*.** In previous studies, it has been shown that mated females of *N. vitripennis* do no longer respond to the male sex pheromone [48, 62] and that this behavioral switch is mediated by the neuromodulator dopamine [10]. Females fed prior to mating with the dopamine receptor antagonist chlorpromazine remained responsive to the male sex pheromone while virgin females injected with a Ringer's solution containing dopamine (injected dosage: 19 ng) became unresponsive [10]. Here, we used the acetone application method to investigate whether the disruptive effect of dopamine on the pheromone response can also be elicited by applying the compound to virgin *N. vitripennis* females. To this end, we produced a stock solution of dopamine in distilled water (2 mg/ml) and diluted this solution 1:10 with acetone. Subsequently, we applied 100 nl of the diluted dopamine solution (representing 20 ng dopamine) dorsally to the region between head and thorax of cold-sedated, virgin females (n = 20). Control females were applied with 100 nl of the pure solvent. Two hours after the application, the responses of these females to the male sex pheromone were tested in a two-choice olfactometer as described in detail elsewhere [10]. Briefly, the residence times of females in two odor fields of a four-chamber olfactometer were recorded for five minutes. One chamber contained a disk of filter paper treated with the male sex pheromone dissolved in dichloromethane (200 ng RS-HDL, 100 ng RR-HDL and 5 ng 4-methylquinazoline), the opposed control chamber contained a solvent-treated filter paper, and the remaining chambers were left empty. The residence times were recorded using The Observer XT.

## Chemical analyses

Chemical analyses were performed using a Shimadzu QP2010 Plus GC/MS system equipped with a 60 m x 0.25 mm inner diameter BPX5 capillary column (film thickness 0.25 μm, SGE Analytical Science Europe, Milton Keynes, UK). Samples (1 μl) were injected splitless at 300˚C using an AOC 20i auto sampler. The MS was operated in the electron impact mode at 70 eV, the mass range was m/z 35–600. Helium was used as carrier gas at a linear velocity of 38 cm/s. The GC program started at 80˚C and was programmed at 5˚C to 300˚C and held at this temperature for 10 min. RS-HDL in the individual samples was quantified by comparing its peak area with the one of the internal standard. For the detection of incorporated $^{13}$C in RS-HDL we checked the extracted ion traces of the diagnostic ions m/z 90, 107, and 120 at the expected retention time of RS-HDL for the appearance of peaks (see 3.5.2) [26, 52]. The incorporation rate was calculated by relating the peak area of a labeled diagnostic ion (m/z 90) to the added peak areas of the respective unlabeled (m/z 86) and the labeled diagnostic ion [24].

**Statistical analyses.** The statistical analyses of the experiments was done using the scientific software PAST 4.03 [66]. Since not all data sets met the assumptions of parametric statistical analysis we used non-parametric methods for data analysis. Longevity, offspring number, and sex ratios as well as the pheromone titers of acetone-treated and control wasps were compared by a Mann-Whitney U-test. The same test was used to compare the time until first mounting, the time from first mounting to the female's receptivity signal, and the copulation durations in the mating trials. The number of observed copulations in these experiments was compared by Fisher's exact test. Pheromone titers in hormone- and acetone-treated control wasps were compared by a Kruskal-Wallis H-test followed multiple Mann-Whitney U-tests with sequential Bonferroni correction. Pheromone responses of females (residence time in pheromone-treated and control fields) were analyzed by a Wilcoxon matched-pairs test.

## Results

### Longevity

The application of acetone to the abdominal tips of cold-sedated, virgin wasps did not influence the longevity of both male and female *N. giraulti* wasps having access to water and honey as a carbohydrate source (Fig 1).

### Mating trials

Acetone treatment of females did not influence the mating behavior of *N. giraulti* couples irrespective of whether mating occurred immediately after application or the next day (Fig 2A–2D). The duration until first mounting, the time from first mounting to the female's receptivity signal, the copulation duration, and the copulation frequency of couples with acetone-treated females did not differ significantly from the control couples. Copulation duration, however, was significantly longer in couples with 1-d old females, irrespective of acetone treatment (Mann-Whitney-U-test: Ac 0d vs. Ac 1d, *p* = 0.03558; Con 0d vs. Con 1d, *p* = 0.0004, Fig 2D).

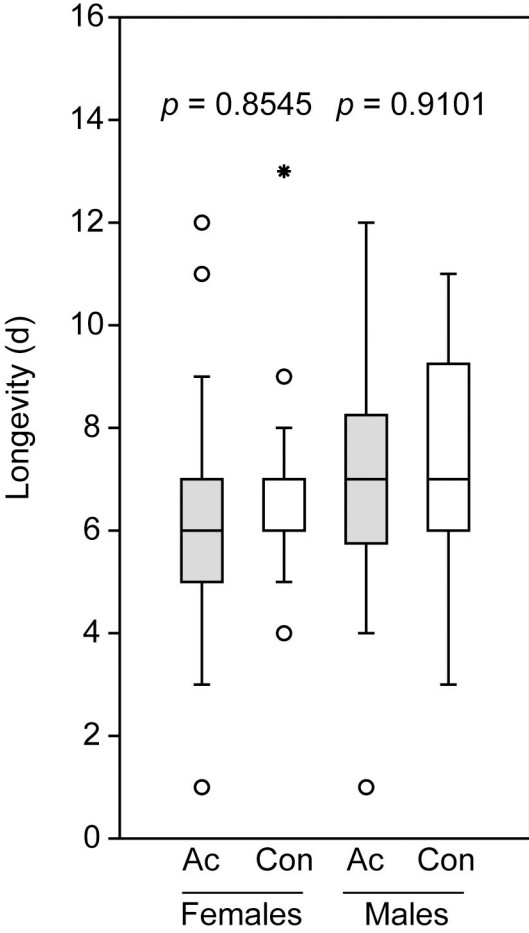

**Fig 1. Longevity of male and female *Nasonia giraulti* treated with acetone (Ac) and untreated control wasps (Con).** Box-and-whisker plots show median (horizontal line), 25–75% quartiles (box), maximum/minimum range (whiskers) and outliers (˚ > 1.5 x above box height; * > 3 x above box height). Data analysis was done by Mann-Whitney U-test, n = 30.

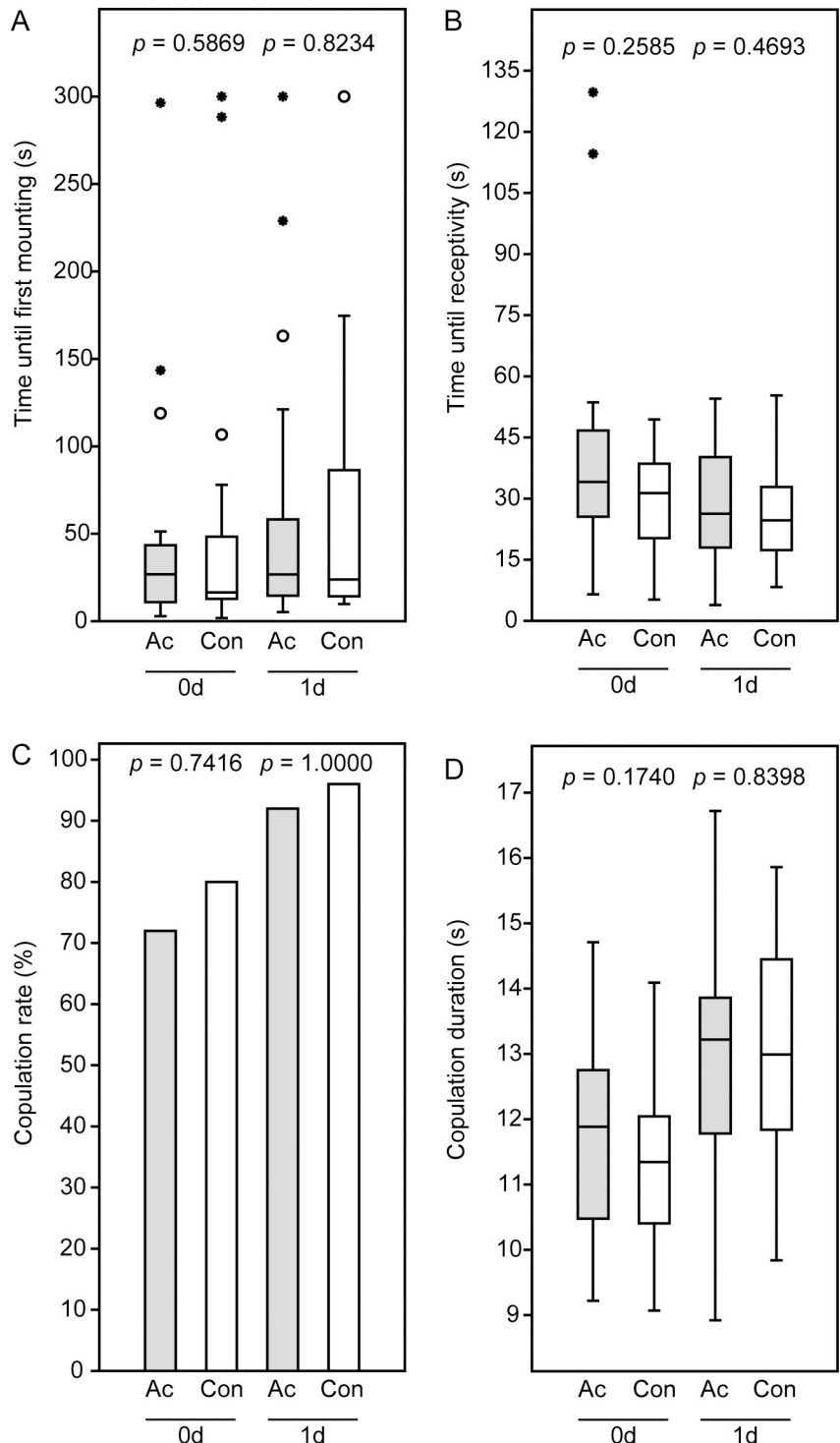

**Fig 2. Results of the mating trials with male and female *Nasonia giraulti*.** Newly emerged virgin females were treated with acetone (Ac) and mated with males either 3–4 h after application (0d) or the next day (1d). Females of the control couples (Con) were subjected to the same procedure without acetone application. (A) Time until males mounted, (B) time between mounting and the female receptivity signal (lowering of the antennae and opening of the genital orifice), (C) copulation rate, and (D) copulation duration. Box-and-whisker plots in panel A, B, and D show median (horizontal line), 25–75% quartiles (box), maximum/minimum range (whiskers) and outliers (° > 1.5 x above box height; * > 3 x above box height). Data analysis by Mann-Whitney U-test (A, B, D) or Fishers exact test (C).

## Fitness and longevity of mated females

Acetone treatment did not influence the longevity of *N. giraulti* females having access to hosts throughout their lifetimes (Fig 3A). This was true for both females mated immediately after acetone treatment and those mated the next day. In comparison to the virgin females provided with a carbohydrate source (Fig 1) but kept in the absence of hosts, however, longevity increased significantly for both acetone-treated and control wasps (acetone: $p < 0.001$, control: $p < 0.001$, Mann-Whitney U-test). Neither offspring number (Fig 3B) nor sex ratio (Fig 3C) were significantly influenced by acetone treatment of females mated immediately after the acetone treatment. In females mated one day after acetone treatment, offspring number was even significantly higher than in the respective controls (Fig 3B) while offspring sex ratio was not influenced (Fig 3C).

## Effect of acetone application on male pheromone titers

Acetone treatment to newly emerged *N. giraulti* males did not influence the pheromone titers (RS-HDL mean ± SEM: 229 ± 48 ng per male) determined two days after the treatment when compared to the control males (226 ± 35 ng per male, Mann-Whitney U-test: $p = 0.5609$) (Fig 4).

## Application examples

**Effect of insect hormones on pheromone titers.** Treatment of newly emerged *N. giraulti* males with 1.04 µg 20-OH-E did not have any significant effect on the pheromone titers of *N.*

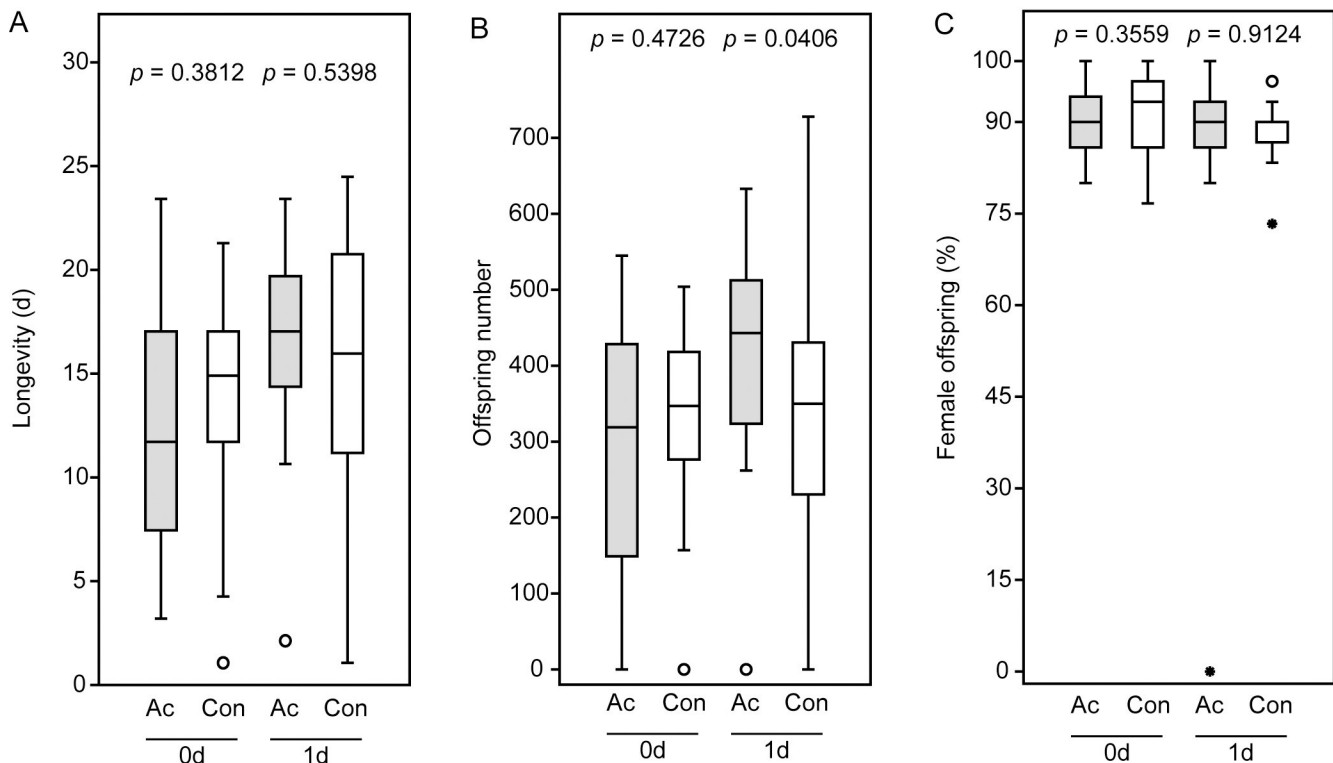

**Fig 3. Fitness parameters of the acetone-treated *Nasonia giraulti* females and untreated control females from the mating trials.** Newly emerged virgin females were treated with acetone (Ac) and mated with males either 3–4 h after application (0d) or the next day (1d). Females of the control couples (Con) were subjected to the same procedure without acetone application. After mating, females were provided with five fresh host pupae per day for oviposition and host feeding until they died. (A) Longevity, (B) offspring number, and (C) offspring sex ratio (given as % female offspring). Box-and-whisker plots show median (horizontal line), 25–75% quartiles (box), maximum/minimum range (whiskers) and outliers (° > 1.5 x above box height; * > 3 x above box height). Data analysis was done by Mann-Whitney U-test.

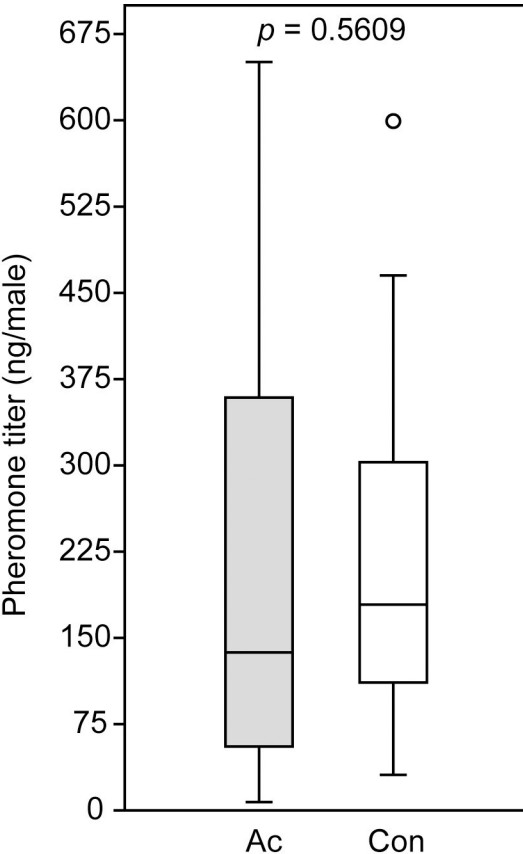

**Fig 4. Pheromone titers of *Nasonia giraulti* males treated with acetone (Ac) and untreated control wasps (Con).** The pheromone component (4*R*,5*S*)-5-hydroxy-4-decanolide was extracted two days after the treatment. Box-and-whisker plots show median (horizontal line), 25–75% quartiles (box), maximum/minimum range (whiskers) and outliers (˚ > 1.5 x above box height above box height. Data analysis was done by Mann-Whitney U-test (n = 20).

*giraulti* males (RS-HDL mean ± SEM: 153 ± 26 ng per male) when compared to the acetone control males (215 ± 130 ng per male). Application of both 1.04 µg or a tenfold reduced dosage of 103.5 ng of JH III, however, reduced the pheromone titers significantly (JH 1000: 24 ± 30 ng per male; JH 100: 36.4 ± 6.6 ng per male; Kruskal-Wallis test: $p < 0.0001$; multiple Mann-Whitney-U-tests with sequential Bonferroni correction: JH 1000 vs. JH 100: $p = 0.0962$; JH 1000 vs. 20-OH-E 1000: $p < 0.0001$, JH 1000 vs. acetone: $p < 0.0001$, JH III100 vs. 20-OH-E 1000: $p = 0.0010$, JH 100 vs. acetone: $p < 0.0001$, 20-OH-E 1000 vs. acetone control: $p = 0.1333$; Fig 5).

**Biosynthesis of the male sex pheromone from linoleic acid.** The diagnostic ions m/z 90, 107, and 120 at the retention time of RS-HDL were detected in all samples from male *N. giraulti* treated with [13]C-labeled linoleic acid (Fig 6) indicating that the labeled fatty acid had been incorporated into the pheromone (mean incorporation rate ± SEM: 6.1 ± 2.2%). Acetone-treated control males did not show peaks at m/z 90 and 120 and a weak signal at m/z 107 which was, however, much weaker than in the [13]C-labeled samples. [13]C-labeling of the diagnostic ions was confirmed by retention times in the respective ion traces that were slightly decreased by ca. 1 s in comparison to the total ion current due to the inverse isotope effect of heavier isotopes on the chromatographic behavior of labeled compounds [67].

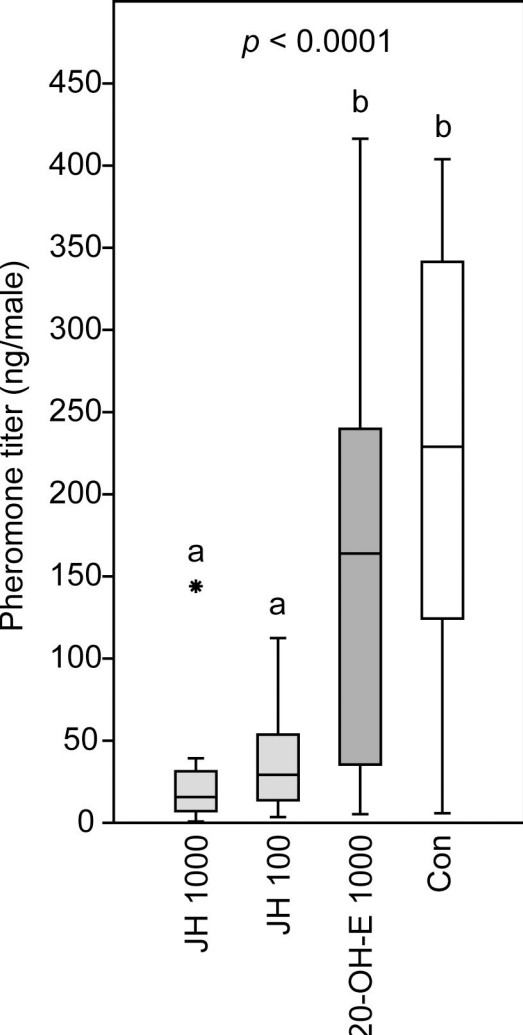

**Fig 5. Effect of insect hormones on the pheromone titers of *Nasonia giraulti* males.** Newly emerged males were treated with acetone solutions containing 1.04 μg or 103.5 ng juvenile hormone III (JH 1000 and JH 100) or 1.04 μg 20-hydroxyecdysone (20-OH-E 1000). Control wasps (Con) were treated with the pure solvent. The pheromone component (4*R*,5*S*)-5-hydroxy-4-decanolide was extracted two days after the treatment. Box-and-whisker plots show median (horizontal line), 25–75% quartiles (box), maximum/minimum range (whiskers) and outliers (˚ > 1.5 x above box height; * > 3 x above box height). Different lowercase letters indicate significant differences at p < 0.05 (data analysis by Kruskal-Wallis ANOVA followed by multiple Mann-Whitney U-tests after sequential Bonferroni correction, n = 20).

### Effect of dopamine on the female sex pheromone response in *Nasonia vitripennis*

Virgin *N. vitripennis* females applied with a dosage of 20 ng dopamine did not prefer the male pheromone in the two-choice bioassay (Wilcoxon matched-pairs test: $p = 0.2322$) whereas virgin control females treated with the pure solvent did prefer the pheromone (Wilcoxon matched-pairs test: $p = 0.0152$, Fig 7).

## Discussion

The present study demonstrates that acetone is a well-suited, non-invasive vehicle to bring bioactive chemicals even into tiny insects such as parasitic wasps. While acetone has been used

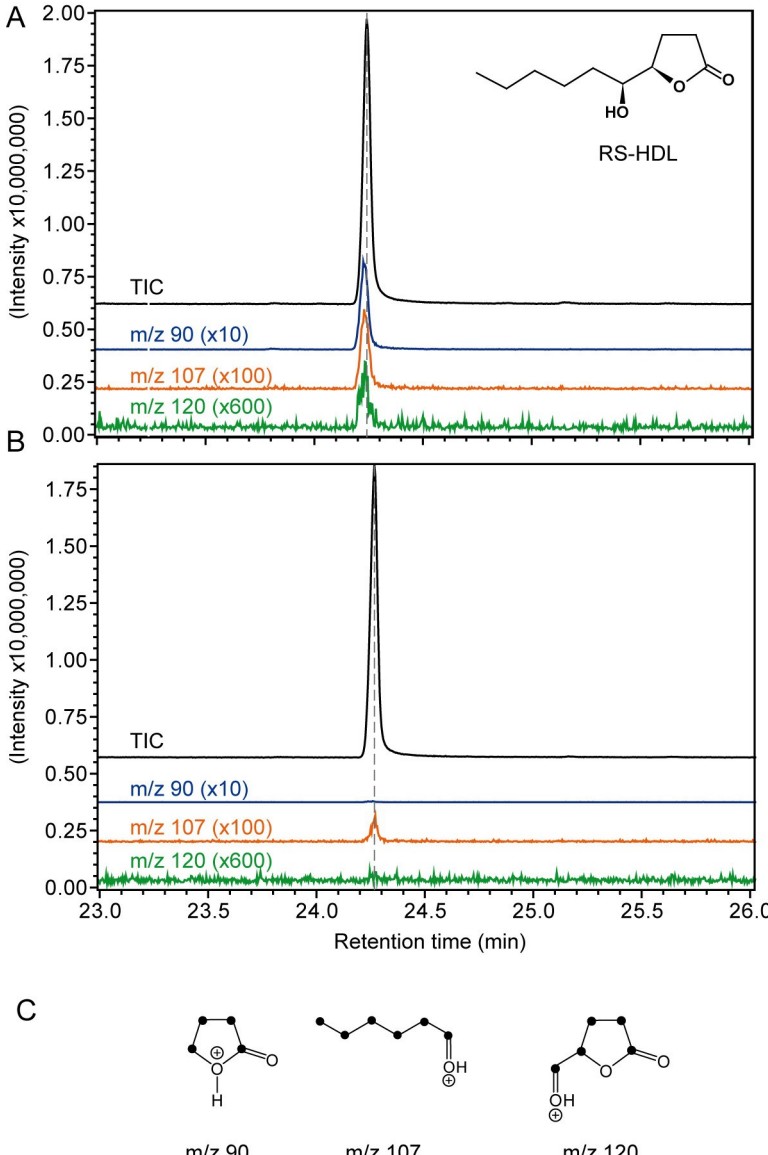

**Fig 6. Results of the ¹³C-labeling experiments.** GC/MS analysis of pheromone extracts from (A) male *Nasonia giraulti* wasps treated with an acetone solution containing fully ¹³C-labeled linoleic acid and (B) control males treated with pure acetone. Shown are the total ion current chromatograms (TIC) and the extracted ion chromatogram of the diagnostic ion m/z 90, 107, and 120 (magnification factors given in parentheses) indicating the incorporation of the ¹³C-labeled precursor into the sex pheromone component (4R,5S)-5-hydroxy-4-decanolide (RS-HDL). The dotted lines indicate the peak maxima of the TIC to illustrate the slightly decreased retention time of the diagnostic ions due to the inverse isotope effect of heavier isotopes. (C) Structures of monitored diagnostic ions (black dots indicate ¹³C-atoms).

previously for this purpose in many studies, a systematic investigation of possible side-effects of the solvent has been rarely performed. At the tested dosage of 207 nl, the application of acetone did neither affect longevity, fecundity, nor sexual communication and mating frequencies of the treated *N. giraulti* wasps. Even though we cannot exclude that certain amounts of acetone evaporate during the application procedure, we assume that the bulk penetrates into the insects via the abdominal intersegmental skin and the anal orifice (see S1 Video). Hence, the solvent, after entering the insect by this way, comes immediately into contact with the internal

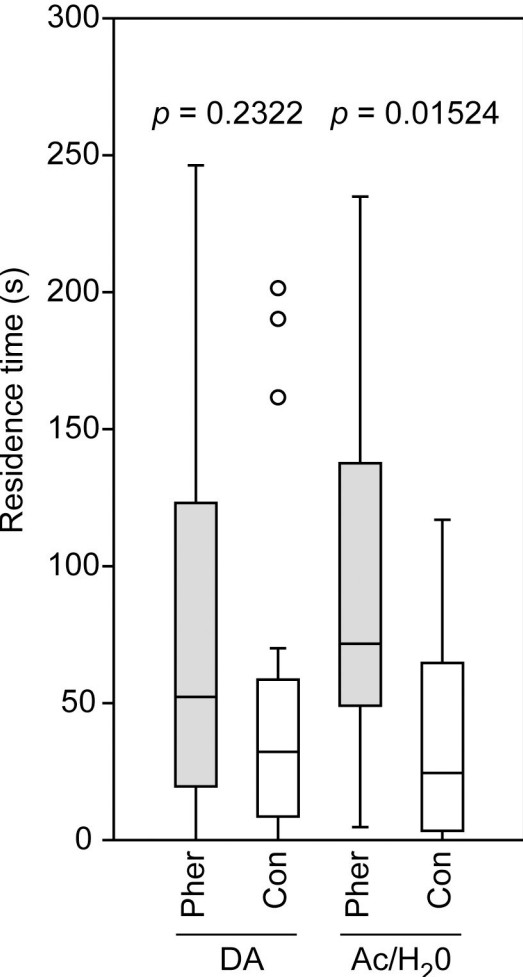

**Fig 7. Effect of dopamine application on the pheromone response of *Nasonia vitripennis* females.** Virgin females were treated with 20 ng dopamine (DA) dissolved in a 1:10 water/acetone mixture ($H_2O/Ac$) or for control with the pure solvent. Given is the residence time of females the test and control areas of a two-choice olfactometer. The test area was equipped with a filter paper disk treated with the male sex pheromone (Pher), the filter paper disk in the control area was treated with the pure solvent (Con). Box-and-whisker plots show median (horizontal line), 25–75% quartiles (box), maximum/minimum range (whiskers) and outliers (° > 1.5 x above box height). Data analysis was done by Wilcoxon matched-pairs test (n = 20).

genitalia of both sexes as well as with the male rectal vesicle, the site where the sex pheromone RS-HDL is produced [26]. Assuming approximately a spherical shape of the wasps' abdomen with a radius of 0.35 mm (the length of the wasps is ca. 2 mm), the volume of the abdomen would be ca.180 nl which is in the range of the applied acetone dosage. Hence, we conclude a high acetone concentration in the abdomen directly after application. Despite this relatively harsh intervention, biomolecules involved in reproduction and pheromone biosynthesis appear to be, if at all, only moderately affected by the solvent. Neither egg nor pheromone production were significantly influenced by acetone application in our experiments. In fact, off-spring number of treated females that were mated one day after the procedure were even significantly increased when compared to the respective control wasps. This suggests that acetone has even positive effects of the biomolecules inside the insects, insects are able to excrete the solvent efficiently or even to metabolize it. For some enzymes, acetone has been shown to

have stabilizing effects even at high concentrations or dosages [68–70]. Metabolization of acetone by carboxylation to acetoacetate has been demonstrated in aerobic and anaerobic bacteria [71, 72]. Acetoacetate in turn is incorporated by rodents and human adipose tissue into fatty acids during lipogenesis [73, 74]. To the best of our knowledge, however, it is unknown whether insects or their symbionts possess the enzymatic machinery to catalyze these reactions and ultimately convert acetone into fatty acids. Whether acetone is used by *Nasonia* wasps to fuel other metabolic pathways needs further investigation.

Previous studies on the acute toxicity of acetone for insects are rare and contradictory. No increased mortality was observed, when acetone was applied topically at dosages of 0.25–2 μl to adult houseflies, *Musca domestica* (Diptera: Muscidae) [32], as well as at 0.5 μl to adult confused flour beetles, *Tribolium confusum* (Coleoptera: Tenebrionidae), or Mediterranean flour moths, *Ephestia kuehniella* (Lepidoptera: Pyralidae) [40]. Application of 1-μl dosages of acetone to ant queens did not influence several fitness parameters but weakly affected hemolymph vitellogenin titers [38]. Acetone has been reported to cause significant mortality in eggs, larvae, and adults of *T. confusum* and *E. kuehniella* when these were exposed to acetone vapor at relatively high concentrations (61.5 and 123 μl/l air) [40]. These results emphasize that the dosage used in application experiments should be kept minimal and adapted to the studied insect species.

The application examples presented in this study emphasize the broad applicability of acetone as a carrier in studies on insect physiology. Topical application of JH III solutions drastically reduced pheromone titers in *N. giraulti* males at both tested dosages of 1.04 μg/wasp and 103.5 ng/wasp. This result is only the first step for a more detailed study addressing the underlying mechanisms. Interestingly, studies on cockroaches [55, 56], beetles [13, 57], and fruit flies [17] found a stimulating effect of juvenile hormones on pheromone biosynthesis, while in moths, JH II seems to act indirectly by priming pheromone glands to respond to the pheromone biosynthesis activating neuropeptide (PBAN) [75].

We demonstrated that the sex pheromone of *N. giraulti* males is produced from linoleic acid. This result confirms previous studies with the congeneric species *N. vitripennis* [52, 53] and demonstrates that acetone application is a suitable and convenient tool to study biosynthetic pathways.

Finally, we demonstrated in this study that topically applied dopamine makes *N. vitripennis* females unresponsive to the male sex attractant. The role of dopamine as neuromodulator in the post-mating behavioral switch has been demonstrated previously by injection and feeding experiments [10].

In toxicological studies, acetone application can be used as a method to investigate also subtler effects than mortality. There is increasing evidence that certain pesticides cause sublethal effects on beneficial insects such as pollinators, predators, or parasitoid wasps [76–78]. Sublethal dosages of the neonicotinoid imidacloprid administered as acetone solutions, for instance, hamper the use of chemical cues and signals during mate and host finding in *N. vitripennis* [21]. Even though solvent control experiments have to be performed routinely for this kind of studies, it is important that the application method itself has the least possible effects on the investigated parameters. Hence, by showing the absence of negative effects of acetone application on both fitness and behavior, the present study lays the foundations for using this technique in toxicological studies addressing sublethal effects of anthropogenic chemicals on insects.

## Supporting information

**S1 Video. Video sequence of the acetone application procedure.** The video shows the application of three doses of 69 nl acetone to the abdomen of a *Nasonia giraulti* wasp.
(M4V)

**S1 Data. Raw data of experimental results.**
(XLSX)

## Acknowledgments

The authors thank Sonja Fleischmann for rearing the insects as well as Nils Schöfer and Gabriel Ratschmann for technical assistance.

## Author Contributions

**Conceptualization:** Joachim Ruther.

**Formal analysis:** Anne-Sophie Jatsch, Joachim Ruther.

**Investigation:** Anne-Sophie Jatsch.

**Methodology:** Joachim Ruther.

**Supervision:** Joachim Ruther.

**Visualization:** Joachim Ruther.

**Writing – original draft:** Joachim Ruther.

**Writing – review & editing:** Anne-Sophie Jatsch.

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
