## [Decision Letter · Decision Letter 0]

18 Dec 2020

PONE-D-20-37248

Acetone application for administration of bioactive substances has no negative effects on longevity, fitness, and sexual communication in a parasitic wasp

PLOS ONE

Dear Dr. Ruther,

Thank you for submitting your manuscript to PLOS ONE. After careful consideration, we invite you to submit a revised version of the manuscript that addresses the points raised during the review process.

We look forward to receiving your revised manuscript.

Kind regards,

Michel Renou, Ph.D

Academic Editor

PLOS ONE

Journal Requirements:

Reviewers' comments:

Reviewer's Responses to Questions

**Comments to the Author**

1. Is the manuscript technically sound, and do the data support the conclusions?

Reviewer #1: Yes

2. Has the statistical analysis been performed appropriately and rigorously? 

Reviewer #1: I Don't Know

3. Have the authors made all data underlying the findings in their manuscript fully available?

Reviewer #1: Yes

4. Is the manuscript presented in an intelligible fashion and written in standard English?

Reviewer #1: Yes

5. Review Comments to the Author

Reviewer #1: This work describes experiments to characterize and assess the effects of acetone, a frequently used solvent vehicle in insect research, on parameters of insect reproduction in parasitic wasps of the genus Nasonia. The work is thorough and presented clearly. It’s refreshing to see solvent toxicity examined with so much attention to detail. Beyond asking how acetone might affect the wasps’ fitness, the researchers also investigate the efficacy of acetone in evaluating the effects of various biomolecules. This is quite informative, and the methods presented could be very useful for future studies involving Nasonia. I have a few comments mostly relating to style and clarity, but also, more details justifying the choice of stats should be added. See below.

Intro

L35-37: Two sentences on the importance of insects to justify studying them feels rushed. Perhaps a few more lines dedicated to describing why insect research in important are merited.

L121-124: More background on the biological questions being addressed by the application experiments in the intro would be beneficial.

Methods

L193: Is this the only component or just the most important sex pheromone? Why was this the only component examined?

L208: 20-OH-E and JH and the research that led to this application test should be discussed in the intro.

L225: The same here for pheromone biosynthesis.

L243: remove “do”

L243: Again, it’s important to mention ahead of the methods why the author chose to perform these experiments.

L280: I am curious why nonparametric approaches were selected. I also wonder what a more inclusive model for certain analysis might yield. For example, if you examine interactive effects of age and treatment, would you see any interactions? A little explanation for the choices made would be helpful.

6. PLOS authors have the option to publish the peer review history of their article (what does this mean?). If published, this will include your full peer review and any attached files.

Reviewer #1: No

---

## [Author Response · Author response to Decision Letter 0]

4 Jan 2021

Answers to each point indicated by ##.

Remark by the editorial office: “We note that you have included the phrase “data not shown” in your manuscript. Unfortunately, this does not meet our data sharing requirements. PLOS does not permit references to inaccessible data. We require that authors provide all relevant data within the paper, Supporting Information files, or in an acceptable, public repository. Please add a citation to support this phrase or upload the data that corresponds with these findings to a stable repository (such as Figshare or Dryad) and provide and URLs, DOIs, or accession numbers that may be used to access these data. Or, if the data are not a core part of the research being presented in your study, we ask that you remove the phrase that refers to these data”

##The phrase was not essential for the paper and therefore removed.##

Reviewer #1: This work describes experiments to characterize and assess the effects of acetone, a frequently used solvent vehicle in insect research, on parameters of insect reproduction in parasitic wasps of the genus Nasonia. The work is thorough and presented clearly. It’s refreshing to see solvent toxicity examined with so much attention to detail. Beyond asking how acetone might affect the wasps’ fitness, the researchers also investigate the efficacy of acetone in evaluating the effects of various biomolecules. This is quite informative, and the methods presented could be very useful for future studies involving Nasonia. I have a few comments mostly relating to style and clarity, but also, more details justifying the choice of stats should be added. See below.

IntroL35-37: Two sentences on the importance of insects to justify studying them feels rushed. Perhaps a few more lines dedicated to describing why insect research in important are merited.

##A few sentences explaining the importance of insects in more detail have been added with some additional references. ##

L121-124: More background on the biological questions being addressed by the application experiments in the intro would be beneficial.

##Some additional sentences introducing the application examples have been added to the introduction.##

MethodsL193: Is this the only component or just the most important sex pheromone? Why was this the only component examined?

##In contrast to N. vitripennis, N. giraulti has only one stereoisomer of 5-hydroxy-4-decanolide in its pheromone (RS-HDL). Information is now given in the introduction. The minor component 4-methylquinazoline was detectable in the samples but not quantifiable by GC/MS.##

L208: 20-OH-E and JH and the research that led to this application test should be discussed in the intro.

##done##

L225: The same here for pheromone biosynthesis.

##done##

L243: remove “do”

##not done##

L243: Again, it’s important to mention ahead of the methods why the author chose to perform these experiments.

##introduction was modified##

L280: I am curious why nonparametric approaches were selected. I also wonder what a more inclusive model for certain analysis might yield. For example, if you examine interactive effects of age and treatment, would you see any interactions? A little explanation for the choices made would be helpful.

##Not all data sets met the assumptions of parametric statistics (normal distribution). That is why we chose non-parametric methods. Information is now given.##

---

## [Editor Report · Decision Letter 1]

6 Jan 2021

Acetone application for administration of bioactive substances has no negative effects on longevity, fitness, and sexual communication in a parasitic wasp

PONE-D-20-37248R1

Dear Dr. Ruther,

We’re pleased to inform you that your manuscript has been judged scientifically suitable for publication and will be formally accepted for publication once it meets all outstanding technical requirements.

Kind regards,

Michel Renou, Ph.D

Academic Editor

PLOS ONE
---

## [Editor Report · Acceptance letter]

11 Jan 2021

PONE-D-20-37248R1 

Acetone application for administration of bioactive substances has no negative effects on longevity, fitness, and sexual communication in a parasitic wasp 

Dear Dr. Ruther:

I'm pleased to inform you that your manuscript has been deemed suitable for publication in PLOS ONE. Congratulations! Your manuscript is now with our production department. 

Kind regards, 

on behalf of

Dr Michel Renou 

Academic Editor

PLOS ONE